

# A new juvenile sauropod specimen from the Middle Jurassic Dongdaqiao Formation of East Tibet

Xianyin An[1], Xing Xu[2,3], Fenglu Han[4], Corwin Sullivan[5,6], Qiyu Wang[1], Yong Li[1], Dongbing Wang[1], Baodi Wang[1] and Jinfeng Hu[4]

[1] Chengdu Center of China Geological Survey, Chengdu, Sichuan, China
[2] Centre for Vertebrate Evolutionary Biology, Yunnan University, Kunming, Yunnan, China
[3] Key Laboratory of Vertebrate Evolution and Human Origins, Institute of Vertebrate Paleontology and Paleoanthropology, China Academy of Sciences, Beijing, China
[4] School of Earth Sciences, China University of Geosciences (Wuhan), Wuhan, Hubei, China
[5] Department of Biological Sciences, University of Alberta, Edmonton, Canada
[6] Philip J. Currie Dinosaur Museum, Wembley, Canada

## ABSTRACT

Jurassic strata are widely distributed in the eastern part of Tibet Autonomous Region, and have yielded many dinosaur bones. However, none of these specimens has been studied extensively, and some remain unprepared. Here we provide a detailed description of some new sauropod material, including several cervical vertebrae and a nearly complete scapula, recovered from the Middle Jurassic of Chaya County, East Tibet. The cervical vertebrae have short centra that bear ventral midline keels, as in many non-neosauropod sauropods such as *Shunosaurus*. Moreover, the cervical centra display deep lateral excavations, partitioned by a septum. The scapula has proximal and distal ends that are both expanded as in mamenchisaurids and neosauropods. However, relatively small body size and lack of fusion of neurocentral sutures in the cervical vertebrae suggest that the available material is from a juvenile, and the length of the cervical centra may have increased relative to the size of the rest of the skeleton in later ontogenetic stages. Phylogenetic analysis provides limited evidence that the new Tibetan sauropod specimen belongs to Eusauropoda, being more derived than *Shunosaurus*, but is basal to Mamenchisauridae. The new material provides important information on the morphological transition between *Shunosaurus* and mamenchisaurids, and extends the known biogeographic range of early-diverging sauropods in the Middle Jurassic of East Asia.

# INTRODUCTION

In Tibet, the highest-altitude region in the world, a series of Jurassic-Cretaceous strata are exposed in the eastern part of Qamdo (Changdu) District. In the 1970s, the Scientific Expedition Team of the Chinese Academy of Sciences discovered many Early-Middle Jurassic dinosaur bones in this area, representing at least ten species and including sauropodomorph, theropod, stegosaur, and early-diverging ornithischian remains (*Zhao,*

Corresponding author
Fenglu Han, hanfl@cug.edu.cn

*1985*; *An et al., 2021*). Almost all these specimens are still unpublished, the sole exception being the partial, medium-sized stegosaur skeleton, comprising the iliosacral region together with two incomplete vertebrae and three dermal plates, that was made the holotype of *Monkonosaurus lawulacus* (*Zhao, 1983*; *Dong, 1990*). However, this species is probably a *nomen dubium* (*Maidment & Wei, 2006*). Some sauropod dinosaur trackways were also reported in the Jurassic of Qamdo, and at least 10 track sites have been discovered (*Xing, Harris & Currie, 2011*; *Xing et al., 2021*). The large pes print length (74–99 cm; *Xing et al., 2021*) and some of the tracks suggest that large sauropods lived in this area during the Early-Middle Jurassic. They may have been closely related to the very abundant sauropods from the Jurassic of the Sichuan Basin, of which about 30 species have been established. Three distinct Jurassic sauropod faunas have been defined within the Sichuan Basin, namely the Early Jurassic *Zizhongosaurus* Fauna, the Middle Jurassic *Shunosaurus-Omeisaurus* Fauna and the Late Jurassic *Mamenchisaurus* Fauna (*Li, 1998*).

In 2019, the field team of the Chengdu Center of the China Geological Survey discovered some new dinosaur fossil sites in Chaya County, Qamdo District, and collected and prepared some sauropod bones (*An et al., 2021*). Here we provide a detailed description of this material and draw comparisons with other sauropods from Gondwana and Laurasia. Our new material may have significant implications for understanding the evolution and diversity of early sauropods in the Jurassic of East Asia.

## MATERIALS & METHODS

The sauropod specimen described in this article (CGS V001) comprises several postcranial elements, which are housed at the Chengdu Center of the China Geological Survey, though only five have so far been prepared. They include four cervical vertebrae and a nearly complete scapula. All these bones were found together within a small area and are likely to be from one individual, though none were preserved in articulation. Field activities were approved by Chengdu Geological Survey Center (project number: DD20190053).

Measurements of the bones are given in Table 1. High-resolution 3D models of the cervical vertebrae and scapula have been uploaded to Morphosource (https://www.morphosource.org/projects/000481873?locale=en). All the bones were scanned using an Artec Space Spider hand-held 3D scanner from China University of Geosciences, and the scans were edited to produce final 3D models using the software Artec Studio.

**Phylogenetic analysis.** To assess the systematic position of the new Tibetan sauropod, we scored it into a recent data matrix for early-diverging sauropods (*Ren et al., 2021*), derived from a previously published matrix (*Xu et al., 2018*). The new matrix contains 386 characters and 77 taxa. Only 30 characters could be scored for the Tibetan sauropod, due to poor preservation. The new matrix was analyzed using TNT v1.5 (*Goloboff & Catalano, 2016*). All characters were treated as equally weighted. 26 characters (12, 58, 95, 96, 102, 106, 108, 115, 116, 119, 120, 145, 152, 163, 213, 216, 232, 233, 234, 235, 252, 256, 298, 299, 301, 379) were treated as ordered, following the original analysis (*Ren et al., 2021*). The maximum number of stored trees was set to 10,000. A New Technology search was performed with default settings, and hit the best score 50 times. The resulting trees were

**Table 1 Measurements of Tibetan sauropod bones.**

| Element | Dimension | Measurement (mm) |
|---|---|---|
| Axis (CGS V001-1) | 1 Centrum length | 126.09 |
| | 2 Anterior centrum height | 92.89 |
| | 3 Anterior centrum width | 67.66 |
| | 4 Centrum height at the mid region | 65.37 |
| | 5 Centrum width at the mid region | N/A |
| | 6 Posterior centrum height | 78.29 |
| | 7 Posterior centrum width | 85.10 |
| | 8 Neural arch length (shortest) | 98.03 |
| | 9 Neural arch height | 84.26 |
| | 10 Neural arch width (anterior end) | 48.12 |
| | 11 Neural canal width (anterior end) | 29.82 |
| | 12 Neural canal height (anterior end) | 45.25 |
| | 13 Neural canal width (posterior end) | 24.52 |
| | 14 Neural canal height (posterior end) | 24.66 |
| | 15 Neural arch width (posterior end) | 45.17 |
| | Ratio of centrum length to posterior centrum height | 1.61 |
| | Ratio of centrum length to posterior centrum width (EI) | 1.48 |
| | Ratio of centrum length to the average of posterior centrum width and height (aEI) | 1.54 |
| Cervical (CGS V001-2) | 1 Centrum length (including ball) | 196.07 |
| | 2 Centrum length (excluding ball) | 153.55 |
| | 3 Anterior condyle height | 71.60 |
| | 4 Anterior condyle width | 86.85 |
| | 5 Anterior centrum height | 78.07 |
| | 6 Anterior centrum width | 110.67 |
| | 7 Centrum height at the mid region | 61.85 |
| | 8 Centrum width at the mid region | N/A |
| | 9 Posterior centrum height | 103.15 |
| | 10 Preserved (estimated) posterior centrum width | 68.35 (105*) |
| | 11 Anterior pleurocoel length | 71.44 |
| | 12 Anterior pleurocoel height | 30.38 |
| | 13 Posterior pleurocoel length | 40.65 |
| | 14 Posterior pleurocoel height | 40.57 |
| | 15 Neural arch length (shortest) | 141.73 |
| | 16 Neural arch height | 79.97 |
| | 17 Neural arch width (mid region) | 83.98 |
| | 18 Neural canal width (anterior end) | 27.40 |
| | 19 Neural canal height | 18.35 |
| | Ratio of centrum length to posterior centrum height | 1.90 |
| | EI value | 1.9[*] |
| | aEI value | 1.9[*] |

**Table 1** (*continued*)

| Element | Dimension | Measurement (mm) |
| --- | --- | --- |
| Cervical (CGS V001-3) | 1 Preserved (estimated) centrum length | 252.06 (300[*]) |
| | 2 Centrum height at the mid region | 125.79 |
| | 3 Posterior centrum height | 123.06 |
| | 4 Preserved (estimated) posterior centrum width | 159.67 (165[*]) |
| | 5 Posterior pleurocoel length | 131.17 |
| | 6 Posterior pleurocoel height | 55.66 |
| | 7 Neural arch length (shortest) | 144.26 |
| | 8 Neural arch height (including neural spine) | 186.55 |
| | 9 Neural spine height | 51.64 |
| | 10 Neural spine width (anteroposterior) | 66.15 |
| | 11 Neural spine thickness (transverse) | 37.46 |
| | 12 Neural canal width (posterior end) | 30.09 |
| | 13 Neural canal height (posterior end) | 30.55 |
| | Ratio of centrum length to posterior centrum height | 2.4[*] |
| | EI value | 1.8[*] |
| | aEI value | 2.1[*] |
| Cervical (CGS V001-4) | 1 Centrum length (including ball) | 263.65 |
| | 2 Centrum length (excluding ball) | 175.57 |
| | 3 Anterior condyle height | 141.64 |
| | 4 Anterior condyle width | 176.65 |
| | 5 Anterior centrum height | 143.00 |
| | 6 Anterior centrum width | 176.93 |
| | 7 Centrum height at the mid region | 107.73 |
| | 8 Centrum width at the mid region | N/A |
| | 9 Posterior centrum height | 156.90 |
| | 10 Posterior centrum width | 160[*] |
| | 11 Right pleurocoel length | 102.02 |
| | 12 Right pleurocoel height | 56.68 |
| | 13 Left pleurocoel length | 117.55 |
| | 14 Left pleurocoel height | 51.94 |
| | 15 Neural arch length (shortest) | 150.17 |
| | 16 Neural arch height | 141.48 |
| | 17 Neural arch width (mid region) | 158.91 |
| | 18 Anterior neural canal width (posterior end) | 30.88 |
| | 19 Anterior neural canal height | 42.26 |
| | 20 Posterior neural canal width (posterior end) | 31.46 |
| | 21 Posterior neural canal height | 33.12 |
| | Ratio of centrum length to posterior centrum height | 1.68 |
| | EI value | 1.65 |
| | aEI value | 1.67 |

**Table 1** (*continued*)

| Element | Dimension | Measurement (mm) |
|---|---|---|
| Scapula (CGS V001-5) | 1 Dorsoventral width of proximal end | 518.44 |
| | 2 Dorsoventral width of mid-region | 186.55 |
| | 3 Preserved dorsoventral width of distal end | 283.83 (incomplete) |
| | 4 Anteroposterior length | 685.97 |
| | 5 Transverse width of proximal end | 51.50 |
| | 6 Transverse width of mid-region | 36.04 |
| | 7 Transverse width of distal end | 61.55 |
| | 8 Transverse width of glenoid rim | 119.60 |
| | 9 Anteroposterior length of glenoid rim | 103.58 |

**Notes.**

*denotes values that have been estimated, based on measurements of the preserved parts of the centrum and comparisons with other cervicals.

subjected to a traditional search using the TBR Swapping algorithm, in order to obtain a final set of most parsimonious trees.

## Geological setting

The specimen described in this article was discovered in Qamdo District, about 10 km from the urban center of Chaya County (Fig. S1). Jurassic strata form an extensively exposed succession in the Qamdo Basin, and mainly comprise lacustrine deposits. The Jurassic strata of the Qamdo Basin include the Lower Jurassic Wangbu Formation, the Middle Jurassic Dongdaqiao Formation, and the Upper Jurassic Xiaosuoka Formation. The new specimen is from the Dongdaqiao Formation, which is about 1.2 km thick and mainly consists of purple red feldspar and quartz-bearing sandstones and siltstones. The dinosaur bones were recovered from red argillaceous siltstone in the middle part of the formation, within a thickness of about 5 m (Fig. 1). The Dongdaqiao Formation has generally been considered to date from the Middle Jurassic, based on its bivalve assemblage (*Wang & Chen, 2005*).

## RESULTS

### Systematic paleontology

Saurischia Seeley, 1887
Sauropodomorpha Huene, 1932
Sauropoda Marsh, 1878
Eusauropoda Upchurch,1995

### Description

Four isolated cervical vertebrae, including an axis, have been prepared. The centra are relatively short, with an average elongation index (aEI: ratio of centrum length to the average of posterior centrum height and width) of about 1.5−2.1 (Table 1). This ratio is similar in *Shunosaurus* (2.0−3.0) from the Middle Jurassic of Sichuan Basin (*Zhang, 1988*), *Tazoudasaurus* (1.6) from the Early Jurassic of Morocco, *Barapasaurus* (about 2) from the Early Jurassic of India (*Allain & Aquesbi, 2008*; *Bandyopadhyay et al., 2010*), and

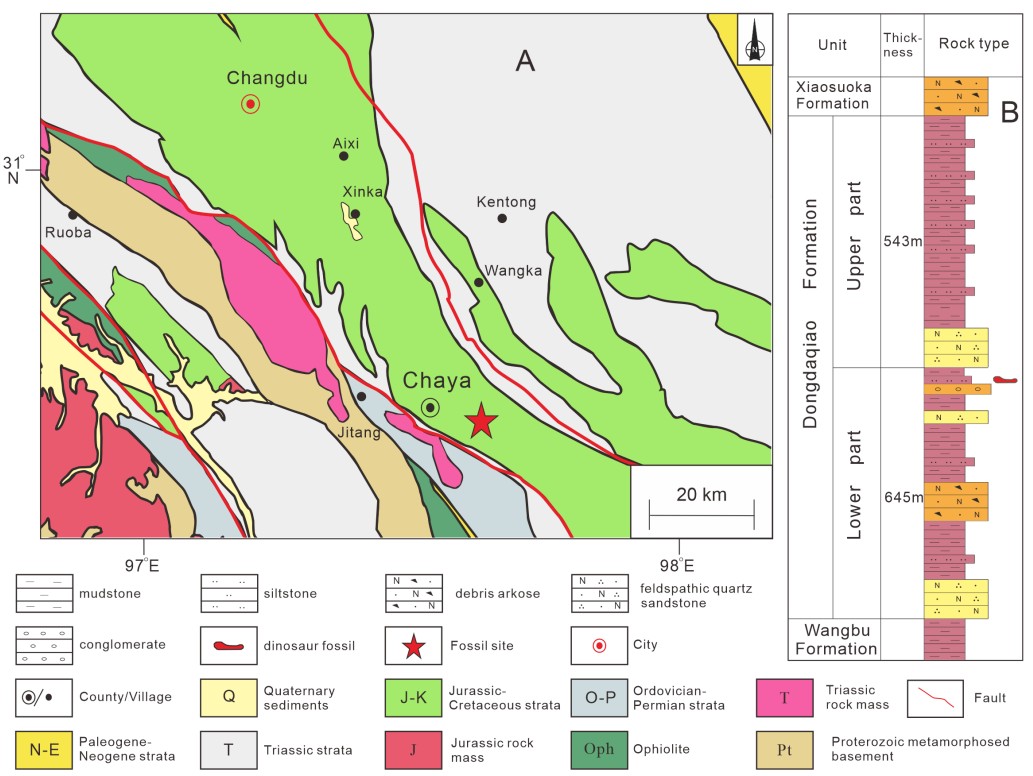

**Figure 1** **Provenance of sauropod remains described in this study.** (A) Location in Chaya County, Qamdo District where the material described in this article was collected; (B) lithostratigraphic column of the Dongdaqiao Formation in the study area.

*Patagosaurus* (1−1.7) from the Middle Jurassic of Argentina (*Holwerda, Rauhut & Pol, 2021*), but larger in *Cetiosaurus* (2.3−2.7), *Bagualia* (3.8−5.3) (*Pol et al., 2020*; *Gomez, Jose & Pol, 2021*; *Holwerda, Rauhut & Pol, 2021*), and mamenchisaurids such as *Analong* (*Ren et al., 2021*) and *Omeisaurus tianfuensis* (1.9−6.1) (*He, Li & Cai, 1988*). The lateral surfaces of the three postaxial cervical centra bear pleurocoels that are partitioned by anterodorsally-trending ridges, as in some cervical vertebrae of *Patagosaurus* (*Holwerda, Rauhut & Pol, 2021*), mamenchisaurids and neosauropods (*Wilson, 2002*). A ventral midline keel is present on the anterior part of each centrum as in the cervical vertebrae of many early-diverging sauropods, such as *Shunosaurus* (*Zhang, 1988*), *Tazoudasaurus* (*Allain & Aquesbi, 2008*), *Omeisaurus* (*He, Li & Cai, 1988*), *Patagosaurus* (*Holwerda, Rauhut & Pol, 2021*), *Bagualia* (*Gomez, Jose & Pol, 2021*), *Spinophorosaurus* (*Remes et al., 2009*), *Lapparentosaurus* (*Upchurch, 1998*), *Amygdalodon* (*Rauhut, 2003*) and an unnamed sauropod from Morocco (*Nicholl, Mannion & Barrett, 2018*), as well as in the anteriormost cervical vertebrae of the Rutland *Cetiosaurus* (*Upchurch & Martin, 2002*), and also dicraeosaurids such as *Lingwulong* (*Xu et al., 2018*). This feature is also present in some non-sauropod sauropodomorphs, including *Yizhousaurus* (*Zhang et al., 2018*), *Massospondylus* (*Barrett et al., 2019*), *Isanosaurus* (*Buffetaut et al., 2000*), and potentially *Antetonitrus* (*McPhee et al., 2014*) and *Lamplughsaura* (*Kutty et al., 2007*).

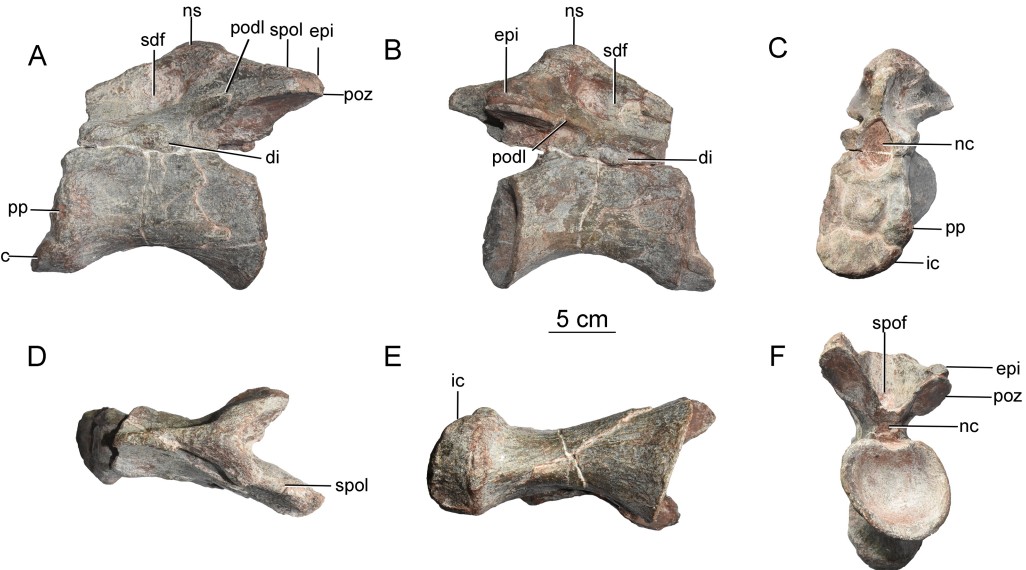

**Figure 2** **Axis of the Tibetan sauropod.** (A) Left lateral view; (B) right lateral view; (C) anterior view; (D) dorsal view; (E) ventral view; (F) posteroventral view. Abbreviations: di, diapophysis; epi, epipophysis; ic, intercentrum; nc, neural canal; ns, neural spine; podl, postzygodiapophyseal lamina; poz, postzygapophysis; pp, parapophysis; sdf, spinodiapophyseal fossa; spof, spinopostzygapophyseal fossa; spol, spinopostzygapophyseal lamina.

**Axis.** The axis (CGS V001-1) is nearly complete, and is well preserved (Fig. 2), with a length of 12.6 cm. The anterior surface of the axis is rugose, bears a pair of dorsoventral grooves, and is tilted to face somewhat dorsally (Fig. 2C). The odontoid process is not preserved. The ventral part of the anterior surface of the centrum contacts, and is fused with, the axial intercentrum (ic) (Fig. 2A). The latter is a small, irregular bone with a crescentic outline in anterior view (Fig. 2C). The ventral surface is smooth and curved dorsally.

The centrum of the axis is relatively elongate (aEI value of 1.5), and transversely compressed. In anterior view, the centrum is taller than wide (Fig. 2C). The posterior surface of the centrum is strongly concave, accommodating the anterior condyle of the third cervical centrum, and has a subcircular outline with equal width and height. The lateral surface of the centrum bears a shallow, elongate fossa with poorly defined margins and no external pneumatic openings, as in the early-diverging eusauropods *Shunosaurus* (*Zhang, 1988*) and *Bagualia* (*Gomez, Jose & Pol, 2021*), whereas the corresponding fossa is deeper in more derived sauropods such as *Omeisaurus* (*He, Li & Cai, 1988*) and *Euhelopus* (*He, Li & Cai, 1988*; *Wilson & Upchurch, 2009*). The fossa is deepest at the anterior end, and gradually becomes shallower posteriorly. The fossa is undivided, as in *Shunosaurus* (*Zhang, 1988*), *Cetiosaurus* (*Upchurch & Martin, 2002*), *Bagualia* (*Gomez, Jose & Pol, 2021*), *Mamenchisaurus hochuanensis* (*Young & Zhao, 1972*) and *Xinjiangtitan* (*Zhang et al., 2020*), whereas the lateral fossa on the axis is partitioned by a ridge in *Omeisaurus* (*He, Li & Cai, 1988*) and more derived sauropods. The parapophysis, positioned on the

anterior margin of the centrum, is a weakly developed structure that takes the form of a convex ridge (Figs. 2A and 2C).

The posterior half of the ventral surface has a gentle transverse convexity. The anterior half of the centrum narrows ventrally but does not form a sharp midline keel of the kind seen in some non-sauropod sauropodomorphs (*e.g.*, *Yizhousaurus* (*Zhang et al., 2018*)) and such early diverging sauropods as *Shunosaurus* (*Zhang, 1988*), *Tazoudasaurus* (*Allain & Aquesbi, 2008*) and *Bagualia* (*Gomez, Jose & Pol, 2021*). The ventral surface of the axis is flat and lacks a midline keel in *Barapasaurus* (*Bandyopadhyay et al., 2010*) and *Mamenchisaurus hochuanensis* (*Young & Zhao, 1972*). In the mamenchisaurid *Xinjiangtitan*, the anterior part of the ventral surface of the axial centrum lacks a keel but bears paired fossae whose outer margins are defined by ventrolateral ridges (*Zhang et al., 2020*).

The neural arch is well developed, but less dorsoventrally tall than the centrum in the mid-region of the vertebra. The part of the neural arch anterior to the apex of the neural spine is taller than the part posterior to the apex (Figs. 2C, 2F). Both diapophyses are largely broken away, but the bases of these structures are anteroposteriorly elongate and located anteroventrally on the neural arch, just above the neurocentral suture (Fig. 2B). No posterior centrodiapophyseal lamina (pcdl) is observable. The anterior opening of the neural canal is large, and taller than wide (Fig. 2C), whereas the posterior opening is relatively small and subcircular, with equal width and height (Fig. 2F; Table 1).

The prezygapophyses are not preserved. The postzygapophyses are large, and extend posterolaterally beyond the posterior part of the centrum. The postzygodiapophyseal lamina (podl) forms a weak ridge extending posterodorsally at an angle of about 30° above the horizontal (Figs. 2A, 2B). A large spinopostzygapophyseal fossa (spof) is present between the postzygapophyses (Fig. 2F), as in *Xinjiangtitan* (*Zhang et al., 2020*). The postzygapophyseal articular facets face ventrally, and are elliptical in outline. The long axis of each facet diverges at 45° from that of the centrum. A prominent epipophysis is clearly present on the dorsal surface of the postzygapophysis (Figs. 2A, 2B, and 2F), as in *Bagualia* (*Gomez, Jose & Pol, 2021*) and *Xinjiangtitan* (*Zhang et al., 2020*). The epipophysis is essentially a dorsal extension of the postzygapophysis, but is separated from the main dorsal surface of the latter by a shallow groove.

The neural spine is weakly developed. The anterior part of the spine is transversely narrow, but a robust laterally projecting ridge extends along the dorsal margin of this portion of the spine and is prominent enough to slightly overhang a deep, distinct fossa (spinodiapophyseal fossa, sdf) situated on the spine's lateral surface, as in the mamenchisaurids *Mamenchisaurus hochuanensis* (*Young & Zhao, 1972*) and *Xinjiangtitan shanshanensis* (*Zhang et al., 2020*). The fossa is slightly deeper than that of *Bagualia* (*Gomez, Jose & Pol, 2021*), whereas in the Rutland *Cetiosaurus* and *Tazoudasaurus* the lateral surface of the neural spine is flattened or convex (*Upchurch & Martin, 2002*; *Allain & Aquesbi, 2008*). The height of the neural spine gradually increases posteriorly, reaching a maximum slightly posterior to the midpoint of the centrum as in *Shunosaurus* (*Zhang, 1988*) and *Mamenchisaurus hochuanensis* (*Young & Zhao, 1972*). In the Rutland *Cetiosaurus* and *Tazoudasaurus*, by contrast, the apex of the neural spine is near the posterior margin of the

centrum (*Upchurch & Martin, 2002*; *Allain & Aquesbi, 2008*). The spinopostzygapophyseal lamina (spol) is straight and robust, and trends ventrolaterally.

**Postaxial cervical vertebrae**. A nearly complete cervical vertebra (CGS V001-2) is well preserved, except that the posterior portion and the left half of the anterior portion of the neural arch are missing, and the left half of the anterior portion of the centrum has likewise been broken away (Fig. 3). The cervical centrum is strongly opisthocoelous, with a prominent hemispherical anterior condyle. The centrum is about 196 mm long, and the ratio of centrum length to posterior centrum height is about 1.9. The posterior articular surface is tilted to face partly ventrally, rather than being perpendicular to the long axis of the centrum. The parapophyses are missing, owing to damage to the anteroventral part of the centrum. The relatively modest height of the preserved neural arch, and the small size of the centrum as a whole, suggest that this vertebra may be from the anterior part of the cervical series.

The lateral surface of the centrum is strongly excavated by a long depression. A thin, sharp, anterodorsally-trending septum (plr) divides the depressed area into two deep pleurocoels (Figs. 3A and 3B). The anterior pleurocoel extends into the centrum in all directions, except posteriorly. The external opening of the anterior pleurocoel is elliptical and anteroposteriorly elongate, whereas the posterior pleurocoel is relatively shallow and subcircular. The ventral surface of the centrum is strongly concave anteroposteriorly, and slightly concave transversely. The ventral surface bears two shallow fossae anteriorly, and a shallow midline keel along its full length.

On the right side of the neural arch, the distal end of the diapophysis is missing, but the basal part of the diapophysis is flattened dorsoventrally, with a thick mid-region and thin anterior and posterior margins. The diapophysis projects laterally and slightly ventrally, and is supported by a well-developed anterior centrodiapophyseal lamina (acdl), which is stout and oriented posterodorsally (Fig. 3A). The partially preserved prezygodiapophyseal lamina (prdl) extends anterodorsally from the diapophysis to the ventrolateral surface of the prezygapophysis (Fig. 3D). The dorsoventrally compressed basal part of the postzygodiapophyseal lamina (podl) is preserved, and extends posterodorsally (Fig. 3D).

The prezygapophyses are broken away, but a stout, vertically aligned centroprezygapophyseal lamina (cprl) is preserved on the right side of the neural arch (Fig. 3E). An intraprezygapophyseal lamina (tprl) extends ventromedially from the prezygapophysis towards the middle of the dorsal edge of the neural canal. The centroprezygapophyseal lamina (cprl) and intraprezygapophyseal lamina (tprl) come together dorsally to define a large, deep fossa. The prezygapophyseal centrodiapophyseal fossa (prcdf) is elliptical, and is situated between the prezygodiapophyseal lamina (prdl) and the anterior centrodiapophyseal lamina (acdl) on the lateral side of the neural arch (Fig. 3A). The postzygapophyses are not preserved, and most of the neural spine is likewise missing. The anterior side of the base of the neural spine is incised by a deep, wide vertical groove (Fig. 3E). A robust spinoprezygapophyseal lamina (sprl) is preserved on the right side, and extends posteriorly, medially and slightly dorsally from the prezygapophyseal area to the neural spine (Fig. 3D).

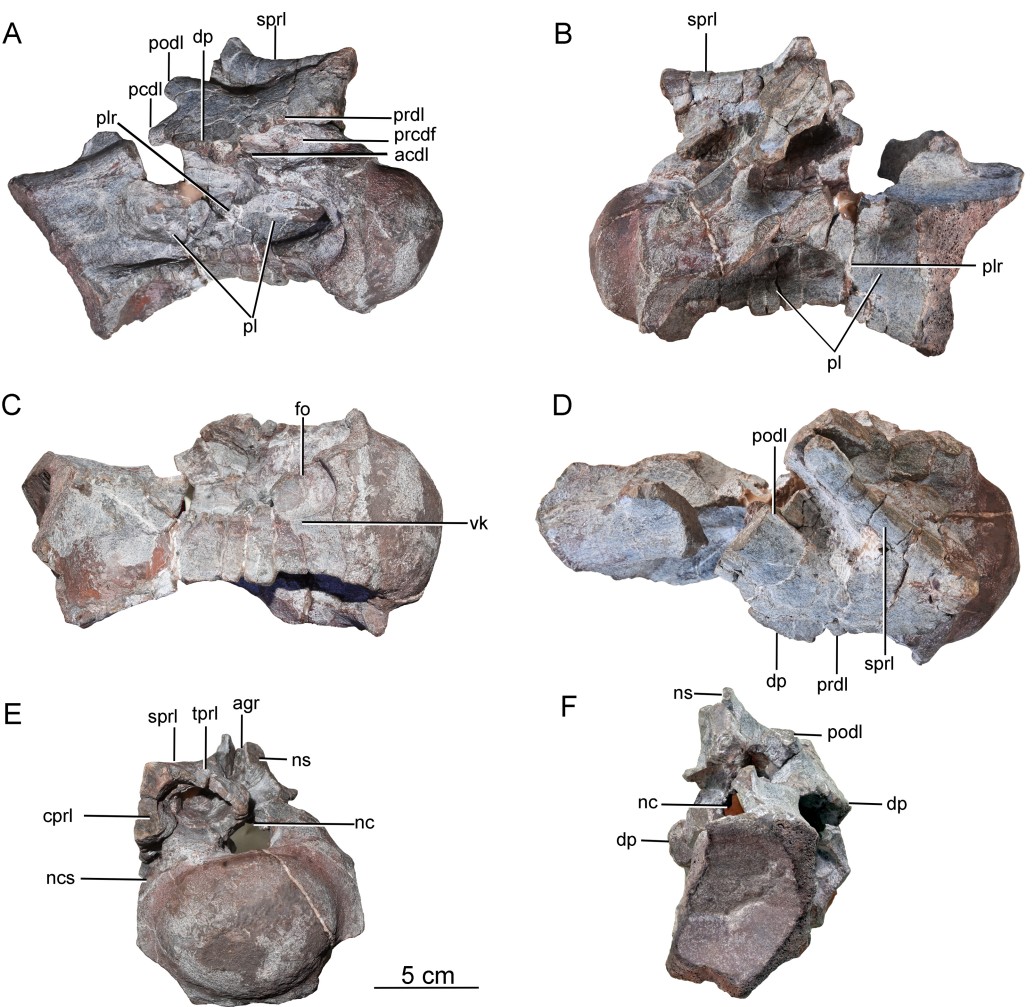

**Figure 3** **Possible anterior cervical vertebra of the Tibetan sauropod.** (A) Right lateral view; (B) left lateral view; (C) ventral view ; (D) dorsal view; (E) anterior view; (F) posterior view. Abbreviations: agr, anterior neural spine groove; acdl, anterior centrodiapophyseal lamina; cprl, centroprezygapophyseal lamina; dp, diapophysis; fo, fossa; tprl, intraprezygapophyseal lamina; nc, neural canal; ncs, neurocentral suture; ns, neural spine; pcdl, posterior centrodiapophyseal lamina; pl, pleurocoel; plr, pleurocoel ridge; podl, postzygodiapophyseal lamina; prcdf, prezygapophyseal centrodiapophyseal fossa; prdl, prezygodiapophyseal lamina; sprl, spinoprezygapophyseal lamina; vk, ventral keel.

A second postaxial cervical vertebra is well preserved, but the anterior part, and much of the right side, of the centrum are missing (CGS V001-3, Fig. 4). This vertebra is relatively large and has a well-developed neural spine, suggesting that it may be from the mid-cervical region. The left lateral surface is excavated by a shallow, elliptical, anteroposteriorly elongate pleurocoel (Fig. 4B). A stout, anterodorsally oriented ridge (plr) forms the pleurocoel's anterior margin. A second pleurocoel was probably originally present anterior to this ridge, as in mamenchisaurid and neosauropod cervical vertebrae. Based on the position of the ridge in typical cervical vertebrae, in fact, it is likely that the anterior half of the centrum is missing. The ventral part of the right half of the centrum is similarly broken away to expose

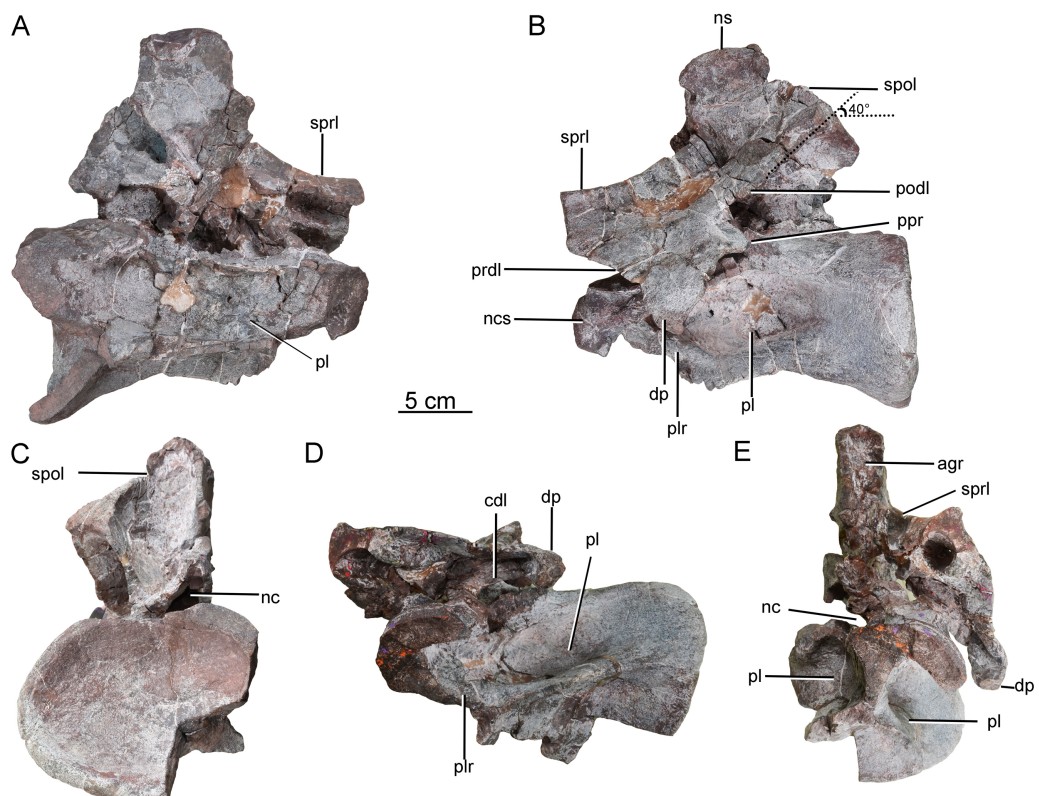

**Figure 4** **Possible mid-cervical vertebra of the Tibetan sauropod.** (A) Right lateral view; (B) left lateral view; (C) posterior view; (D) ventral view; (E) anterior view. Abbreviations: agr, anterior neural spine groove; cdl, centrodiapophyseal lamina; dp, diapophysis; nc, neural canal; ncs, neurocentral suture; ns, neural spine; pl, pleurocoel; plr, pleurocoel ridge; podl, postzygodiapophyseal lamina; ppr, posterior process; prdl, prezygodiapophyseal lamina; spol, spinopostzygapophyseal lamina; sprl, spinoprezygapophyseal lamina.

the centrum's internal structure. The large, anteroposteriorly elongate pleurocoel (Fig. 4A) resembles that of the early-diverging macronarian *Camarasaurus* (*Wedel, 2003*), but lacks the camellate internal cavities that occur in derived titanosaurs. The preserved part of the ventral surface is strongly concave both transversely and anteroposteriorly. The preserved part of the centrum lacks a prominent ventral keel, but a ventral keel may have been present more anteriorly, as in the cervical vertebrae of other early-diverging sauropods and some massopodans.

On the left side of the neural arch, the diapophysis is well preserved, has a subtriangular outline in dorsal view, and tapers ventrolaterally (Fig. 4B). The dorsal surface of the diapophysis is flattened. The prezygodiapophyseal lamina (prdl) is partially preserved as a sheet of bone arising from the base of the anterior edge of the diapophysis, with a thin edge that extends anterodorsally (Fig. 4B). The postzygodiapophyseal lamina (podl) extends from the base of the diapophysis to the lateral margin of the postzygapophysis, forming an angle of about 40° with the long axis of the centrum (Fig. 4B). A prominent, tapering process protrudes posteroventrally from the base of the diapophysis (Fig. 4B: ppr),

resembling the costal spurs present in the neosauropod *Euhelopus zdanskyi*. However, the costal spurs of *Euhelopus* are less prominent and more distally located (*Wilson & Upchurch, 2009*).

Both the pre- and postzygapophyses are missing. The spinoprezygapophyseal lamina (sprl) is sharp, its margin curving posterodorsally from the prezygapophyseal area to merge with the anterior edge of the neural spine (Fig. 4B). The neural spine is well preserved, subrectangular in outline in lateral view, and transversely compressed. The anterior neural spine groove is present and transversely narrow. In posterior view, the spinopostzygapophyseal fossa (spof) is deep and tall (Fig. 4C). The neural canal is subcircular and much smaller than the posterior surface of the centrum.

A third postaxial cervical vertebra can be recognized as a posterior member of the cervical series, based on the relative shortness of the centrum and pleurocoel (ratio of centrum length to posterior centrum height of only about 1.68) (CGS V001-4, Fig. 5). The centrum is strongly opisthocoelous, with a prominent hemispherical anterior condyle. The lateral surface of the centrum is strongly excavated by a deep, elliptical pleurocoel (Fig. 5A). The right parapophysis is broken away, but the left parapophysis is located on the anteroventral region of the centrum and tapers laterally, having a triangular outline in lateral and anterior views (Fig. 5B).

The ventral surface is strongly concave anteroposteriorly, the apex of the concavity being located in the anterior half of the centrum. A strong ventral midline keel is present, and extends along the entire length of the centrum. The midline keel is sharp and deep anteriorly, and progressively becomes wider and less prominent towards the centrum's posterior end (Fig. 5D). The centroprezygapophyseal lamina (cprl) is a simple stout ridge, extending anterodorsally from the diapophysis to support the prezygapophysis (Fig. 5C).

The left prezygapophysis is well preserved, with a facet that faces dorsomedially and is teardrop-shaped, tapering posteromedially to a point. The articular surface is flattened. A large, shallow fossa is present on the underside of the prezygapophysis (Fig. 5C). The spinoprezygapophyseal lamina (sprl) forms a prominent ridge extending posterodorsally from the prezygapophysis (Fig. 5A). The diapophysis and the posterior part of the prezygodiapophyseal lamina (prdl) are broken away. The anterior part of the prezygodiapophyseal lamina (prdl) is preserved as a stout ridge, whereas the postzygodiapophyseal lamina (podl) appears thinner and more sheet-like, based on the preserved basal part of the latter (Figs. 5B, 5E). The anterior centrodiapophyseal lamina (acdl) is thick and extends posterodorsally, and the posterior centrodiapophyseal lamina (pcdl) extends anterodorsally. Together with the dorsal lamina on the centrum (dlc), they define a deep fossa situated ventral to the diapophysis and visible in lateral view (Fig. 5B) as in the posterior cervical vertebrae of *Europasaurus* (*Carballido & Sander, 2014*). The prezygapophyseal centrodiapophyseal fossa (prcdf) is deep and narrow anteroposteriorly (Fig. 5B). Both postzygapophyses are missing, as is the neural spine. The lateral centropostzygapophyseal lamina (lcpol) is robust and vertically directed.

**Scapula.** The left scapula is nearly complete, lacking only small portions of the proximal plate and distal expansion (CGS V001-5, Fig. 6), and is flat and elongate. The lateral and medial surfaces of the proximal plate are both shallowly excavated, but

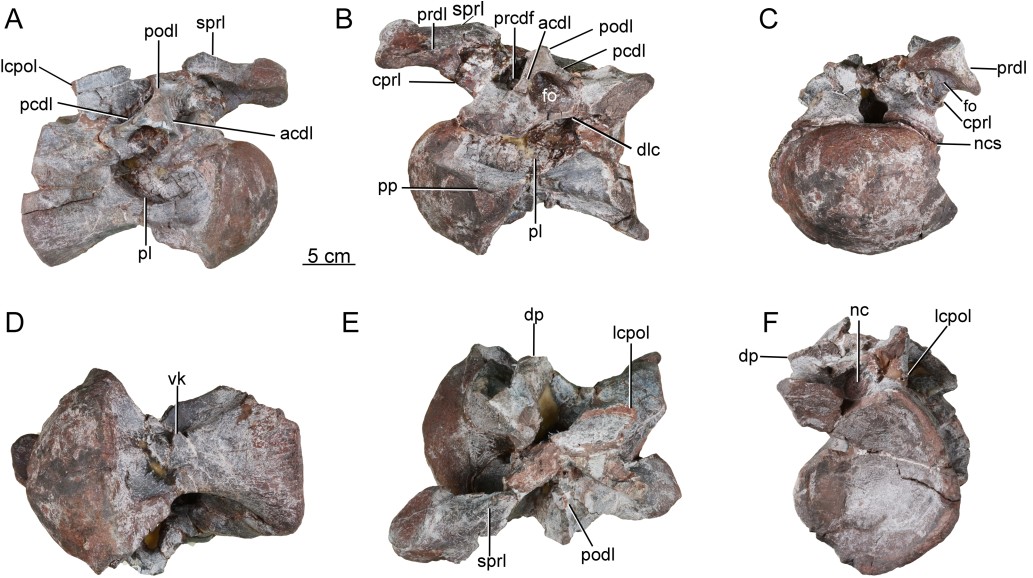

**Figure 5** **Possible posterior cervical vertebra of the Tibetan sauropod.** (A) Right lateral view; (B) left lateral view; (C) anterior view; (D) ventral view; (E) dorsal view; (F) posterior view. Abbreviations: acdl, anterior centrodiapophyseal lamina; cprl, centroprezygapophyseal lamina; dlc, dorsal lamina of the centrum; dp, diapophysis; fo, fossa; lcpol, lateral centropostzygapophyseal lamina ; nc, neural canal; ncs, neurocentral synostosis; pcdl, posterior centrodiapophyseal lamina; pl, pleurocoel; podl, postzygodiapophyseal lamina; pp, parapophysis; prcdf, prezygapophyseal centrodiapophyseal fossa; prdl, prezygodiapophyseal lamina; sprl, spinoprezygapophyseal lamina; vk, ventral keel.

the acromial ridge that is present in most neosauropods (*Upchurch, Barrett & Dodson, 2004*) is lacking. The dorsoventral height of the strongly expanded proximal plate is estimated to be more than 50% of the total length of the scapula (about 0.6), as in mamenchisaurids and more advanced sauropods (*Upchurch, Barrett & Dodson, 2004*). The acromial process is moderately developed and its posterior margin is slightly convex, which is similar to the condition in *Lingwulong* (*Xu et al., 2018*), *Lapparentosaurus*, the Rutland *Cetiosaurus* and *Patagosaurus* (*Upchurch & Martin, 2002*; *Holwerda, 2019*). Comparatively, the acromial process is better developed in mamenchisaurids (Fig. 7), but poorly developed in *Shunosaurus* (*Zhang, 1988*). The long, anteroventrally protruding glenoid region is transversely thick. The glenoid region is rectangular in lateral view, and bears a slightly rugose articular surface. The lateral and medial surfaces of the scapular blade are both convex, creating a lenticular cross section, although the convexity of the lateral surface is more pronounced. The blade is slightly deflected medially, relative to the proximal plate. The distal end of the blade is strongly expanded dorsoventrally, though the dorsal part of the expanded area is slightly damaged. The distal end of the scapular blade is also strongly expanded in *Omeisaurus*, *Mamenchisaurus*, *Yuanmousaurus* (*Lu et al., 2006*), the Rutland *Cetiosaurus* (*Upchurch & Martin, 2002*), and *Patagosaurus* (*Holwerda, 2019*), but only slightly expanded in the early-diverging sauropods *Shunosaurus* (*Zhang, 1988*) and *Barapasaurus* (Bandyopadhyay et al., 2010), and in the diplodocoid *Lingwulong* (Fig. 7).

none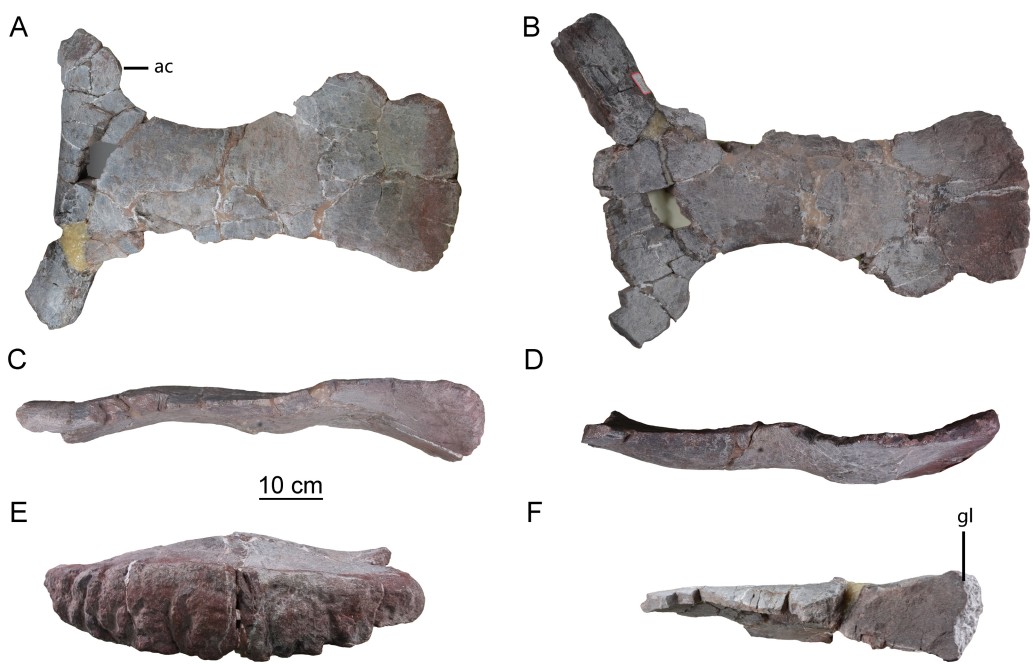

**Figure 6  Left scapula of the Tibetan sauropod.** (A) Lateral view; (B) medial view; (C) ventral view; (D) dorsal view; (E) posterior view; (F) anterior view. Abbreviations: ac, acromial process; gl, glenoid.
Full-size 🖼 DOI: 10.7717/peerj.14982/fig-6

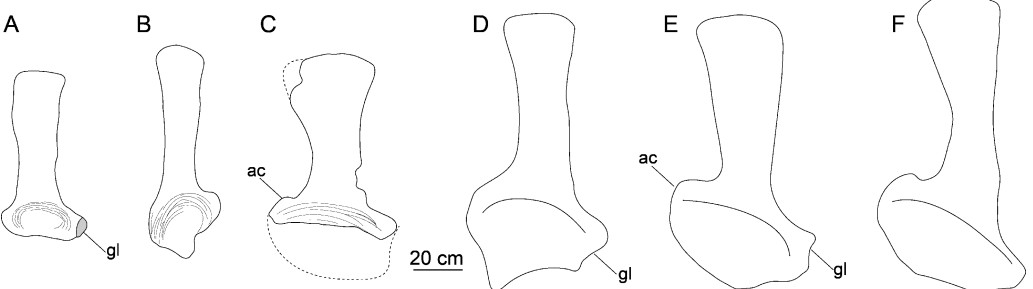

**Figure 7  Comparison of left scapulae in lateral view.** (A) *Tonganosaurus hei* (MCDUT 14454, reversed); (B) *Shunosaurus lii* (ZDM T 5402); (C) Tibetan sauropod (CGS V001); (D) *Lingwulong shenqi* (LM V001b, reversed); (E) *Omeisaurus tianfuensis* (ZDM T5704); (F) *Mamenchisaurus youngi* (ZDM0083). Abbreviations: ac, acromial process; gl, glenoid.

## DISCUSSION

The recovered bones of the new Tibetan sauropod dinosaur are generally similar to those of other Early and Middle Jurassic sauropods, and also preserve some derived features previously known in mamenchisaurids. The cervical vertebrae are opisthocoelous and short, as in the Early Jurassic sauropods *Kotasaurus* from India (*Yadagiri, 2001*), *Tazoudasaurus* from Morocco (*Allain & Aquesbi, 2008*), and *Zizhongosaurus* and *Gongxiansaurus* (which can now be studied only on the basis of information in the literature, because the only

known specimen may have been destroyed in the collapse of the exhibition hall in which it was displayed) from the Sichuan Basin (*He et al., 1998*; *Xing et al., 2019*), as well as the middle Jurassic *Shunosaurus* (*Zhang, 1988*) and *Patagosaurus* (*Holwerda, Rauhut & Pol, 2021*). The lateral surfaces of the centra are excavated as in most sauropods, such as *Tonganosaurus* from the Lower Jurassic Yimen Formation of the Sichuan Basin (*Li et al., 2010*), but the cervical vertebrae of *Tonganosaurus* are more elongated and have no septa in their lateral excavations. The shallow concavity of the lateral surface of the axial centrum, together with the lateral excavations and ventral midline keels on the postaxial cervical centra, represent strong similarities to Middle Jurassic sauropods from the Sichuan Basin, such as *Shunosaurus* and *Dashanpusaurus* (*Zhang, 1988*; *Peng et al., 2005*), and also to *Patagosaurus* (*Holwerda, Rauhut & Pol, 2021*). In addition, the Tibetan sauropod bones display some features seen in mamenchisaurids and neosauropods, such as a relatively robust scapula with a strongly dorsoventrally expanded proximal plate. However, the Tibetan sauropod also lacks many derived features of mamenchisaurids, including a deep lateral excavation on the axis, elongated cervical vertebrae, cervical centra with three or more lateral excavations and no ventral midline keel, and bifurcate cervical neural spines (*Young & Zhao, 1972*; *He, Li & Cai, 1988*; *Ouyang & Ye, 2002*; *Ren et al., 2021*).

Our phylogenetic analysis resulted in 78 most parsimonious trees with a length of 1223 (consistency index equals 0.373; retention index equals 0.702). The strict consensus tree supports referral of the Tibetan sauropod to Sauropoda (Fig. S2), but a reduced consensus tree indicates that the Tibetan sauropod is the most positionally unstable OTU. The majority-rule consensus tree posits the Tibetan sauropod as a eusauropodan more derived than *Shunosaurus*, but excludes it from Mamenchisauridae and Neosauropoda (Fig. 8).

It is difficult to ascribe the Tibetan sauropod specimen to any known sauropod species or genus. The shortness of the cervical vertebrae resembles the condition in *Shunosaurus*, but the vertebrae bear more complicated excavations than are present in that taxon. However, the complexity of the cervical excavations may be subject to ontogenetic variation in sauropods. While documented examples of ontogenetically-driven morphological changes in sauropods are scant, such changes have been reported in a few genera, including *Shunosaurus* (*Ma et al., 2022*), *Brachiosaurus* (*Carballido et al., 2012*), *Europasaurus* (*Carballido & Sander, 2014*) and *Barosaurus* (*Melstrom et al., 2016*). Information from these taxa implies that the pleurocoels of the cervical and dorsal vertebrae became more structurally complex, and the cervical centra more elongate, in older individuals. In particular, *Carballido & Sander (2014)* divided the ontogeny of *Europasaurus* into five stages, based on the degree to which pleurocoels and laminae were developed.

The new Tibetan sauropod specimen may be a juvenile, based on its relatively small size and the presence of visible neurocentral sutures (ncs, Figs. 3E, 4B, 5C). The axial centrum is about as long (126 mm) as a complete example of the same element in a *Shunosaurus* specimen (125 mm, ZDM T5042) which was estimated to have had a total body length of 11 m (*Zhang, 1988*). The maximum length of the preserved scapula is estimated to be less than 70 cm (based on the scapular proportions of mamenchisaurids), making it much shorter than the scapulae of adult individuals of such early-diverging eusauropod taxa as *Shunosaurus* (90 cm, ZDM T5042) (*Zhang, 1988*) and *M. youngi* (119 cm, ZDM0083)

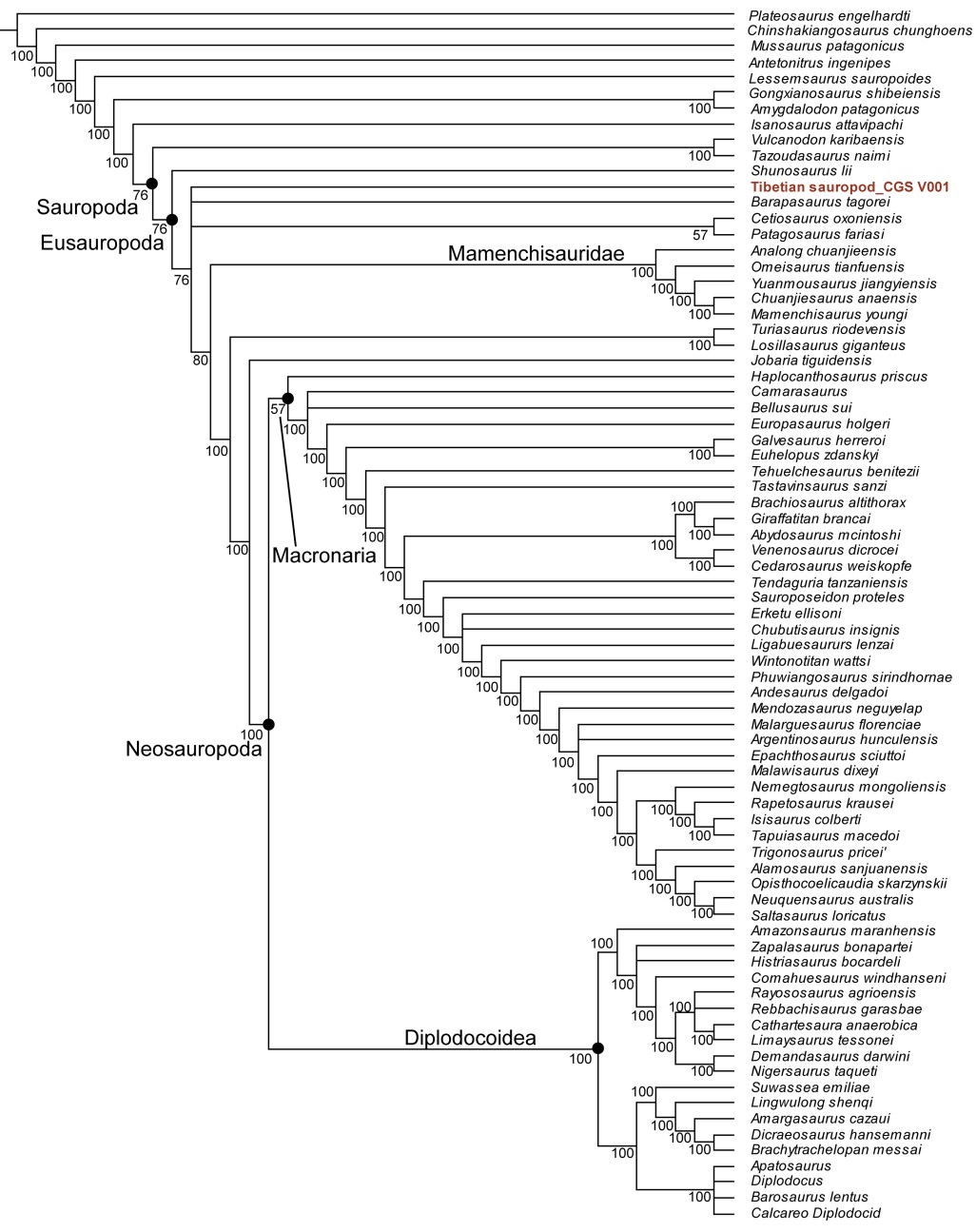

**Figure 8 Majority rule consensus tree from 78 MPTs (TL = 1,223), positing the Tibetan sauropod as more derived than _Shunosaurus_ but basal to Mamenchisauridae.**

(_Ouyang & Ye, 2002_). However, the scapula of the Tibetan sauropod is slightly larger than that of a recently described partial juvenile _Shunosaurus_ skeleton (scapula length 57.4 cm) from the Middle Jurassic of Chongqing Municipality, China (CLGPR V00007) (_Ma et al., 2022_). The latter has a slender shaft and low acromial process as in adults of _Shunosaurus_, and in contrast to the condition in the Tibetan sauropod. Unfortunately, no cervical vertebrae are preserved in the juvenile _Shunosaurus_, and most of the ontogenetic

variations that could be inferred based on this specimen pertained to limb bones that are not represented in the Tibetan material (*Ma et al., 2022*).

Nevertheless, the lack of fusion of the neurocentral sutures in the preserved cervical vertebrae suggests the new Tibetan sauropod material represents a juvenile individual. Using the criteria established by *Carballido & Sander (2014)* (and assuming that the ontogeny of *Europasaurus* was similar to that of the presumably much more ancestral taxon represented by the Tibetan material), the new Tibetan sauropod specimen can be recognized as a "late immature" individual, as well-developed laminae and fossae are apparent in the cervical series but the cervical vertebrae are still short. Similarly, the juvenile holotype of *Daanosaurus* from the Upper Jurassic of the Sichuan Basin has a very short axis (*Ye, Gao & Jiang, 2005*), and juvenile specimens of *Bellusaurus* from the Middle Jurassic of the Xinjiang Autonomous Region have relatively short cervical vertebrae that bear deep excavations divided by septa, although in the *Bellusaurus* material the cervical vertebrae lack ventral midline keels and the scapulae are relatively slender (*Dong, 1990*). These comparisons indicate that the cervical vertebrae of the Tibetan sauropod would likely have developed more complex excavations and increased in size relative to other parts of the skeleton if ontogeny had continued, suggesting that an adult of the same species would have been more similar to mamenchisaurids (*Carballido & Sander, 2014*).

To summarize, the new Tibetan sauropod specimen displays a unique combination of features not seen in other early-diverging sauropods. However, more material is needed before a new taxon can be established, due to the incompleteness of the preserved bones and their juvenile status. The similarities between the Tibetan specimen and mamenchisaurids, which are already known to have a wide distribution in the Middle Jurassic of Asia (*Ren et al., 2021*), suggest that the Tibetan specimen may be at least closely related to Mamenchisauridae, particularly when possible ontogenetic effects are taken into account. A detailed study of ontogenetic variation in mamenchisaurids would be helpful in more confidently establishing the taxonomic position of the Tibetan specimen.

## CONCLUSIONS

The Tibetan sauropod bones reveal the presence of a short-necked early-diverging sauropod in the Middle Jurassic Dongdaqiao Formation of Chaya County, Qamdo District. Among previously described taxa, the specimen is most closely similar to early-diverging eusauropods from the Middle Jurassic, the resemblances including the shortness of the cervical centra and the fact that they bear lateral excavations. The specimen also possesses some derived features seen in the Late Jurassic mamenchisaurids and neosauropods, such as a robust scapula with a strongly dorsoventrally expanded proximal end, and a deep fossa on the lateral surface of the axial neural spine. The small size of the available bones and the visible neurocentral sutures on the preserved cervical vertebrae suggest that the specimen represents a juvenile, which might have increased in relative neck length and the complexity of the pleurocoels and laminae in the cervical region if growth had continued. Therefore, an adult individual of the same species might show clearer similarities to mamenchisaurids. The new material provides significant information on the morphological transition from

early-diverging eusauropods to mamenchisaurids, and expands the known diversity and biogeographic range of sauropods in the Middle Jurassic of East Asia.

**Institutional abbreviations**

| | |
|---|---|
| **CLGPR** | Chongqing Laboratory of Geoheritage Protection and Research |
| **LM** | Lingwu Museum |
| **ZDM** | Zigong Museum |
| **MCDUT** | Museum of Chengdu University of Technology |

## ACKNOWLEDGEMENTS

We thank field team members Tao Yang and Yucong Ma for collecting the fossils described in this article, Xiaobing Wang from Tianyan Museum for preparing the fossils, and Xuefang Wei for very useful discussion. We thank the editor, Emanuel Tschopp, as well as Omar Rafael Regalado Fermandez, Femke Holwerda and an anonymous reviewer, for their very helpful comments on this manuscript.

### Funding

This project was supported by the National Natural Science Foundation of China (grant numbers 41602126; 42288201; 41972021), the China Geological Survey (DD20190053, DD20221811, DD20221661, DD20221645), the Second Tibetan Plateau Scientific Expedition and Research Program (grant 2019QZKK0706), the National Sciences and Engineering Research Council of Canada (Discovery Grant RGPIN-2017-06246), and start-up funding awarded by the University of Alberta. The funders had no role in study design, data collection and analysis, decision to publish, or preparation of the manuscript.

### Grant Disclosures

The following grant information was disclosed by the authors:
The National Natural Science Foundation of China: 41602126, 42288201, 41972021.
The China Geological Survey: DD20190053, DD20221811, DD20221661, DD20221645.
The Second Tibetan Plateau Scientific Expedition and Research Program: 2019QZKK0706.
The National Sciences and Engineering Research Council of Canada: RGPIN-2017-06246.
The University of Alberta.

### Competing Interests

The authors declare there are no competing interests.

### Author Contributions

- Xianyin An conceived and designed the experiments, performed the experiments, analyzed the data, prepared figures and/or tables, authored or reviewed drafts of the article, and approved the final draft.
- Xing Xu conceived and designed the experiments, performed the experiments, analyzed the data, authored or reviewed drafts of the article, and approved the final draft.

- Fenglu Han conceived and designed the experiments, performed the experiments, analyzed the data, prepared figures and/or tables, authored or reviewed drafts of the article, and approved the final draft.
- Corwin Sullivan conceived and designed the experiments, performed the experiments, analyzed the data, authored or reviewed drafts of the article, and approved the final draft.
- Qiyu Wang conceived and designed the experiments, authored or reviewed drafts of the article, and approved the final draft.
- Yong Li conceived and designed the experiments, authored or reviewed drafts of the article, and approved the final draft.
- Dongbing Wang conceived and designed the experiments, authored or reviewed drafts of the article, and approved the final draft.
- Baodi Wang conceived and designed the experiments, authored or reviewed drafts of the article, and approved the final draft.
- Jinfeng Hu conceived and designed the experiments, authored or reviewed drafts of the article, and approved the final draft.

## Field Study Permissions

The following information was supplied relating to field study approvals (i.e., approving body and any reference numbers):

Field experiments were approved by Chengdu Geological Survey Center (Project number: DD20190053).

## Data Availability

Fossil site and phylogenetic trees are available in the Supplemental Files. Raw measurements are available in Table 1.

All specimens are stored in Chengdu Center of the China Geological Survey. Accession numbers: CGS V001-1, CGS V001-2, CGS V001-3, CGS V001-4, CGS V001-5.

High-resolution 3D models of the cervical vertebrae and scapula are available at Morphosource: https://www.morphosource.org/projects/000481873?locale=en
  - CGS V001-1: https://doi.org/10.17602/M2/M481877
  - CGS V001-2: https://doi.org/10.17602/M2/M482213
  - CGS V001-3: https://doi.org/10.17602/M2/M482218
  - CGS V001-4: https://doi.org/10.17602/M2/M482223
  - CGS V001-5: https://doi.org/10.17602/M2/M482228

## Supplemental Information

Supplemental information for this article can be found online at http://dx.doi.org/10.7717/peerj.14982#supplemental-information.

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
