# Peer review of "A new juvenile sauropod specimen from the Middle Jurassic Dongdaqiao Formation of East Tibet"

_PeerJ, doi:10.7717/peerj.14982_

## Round 0.1 · original submission · Minor Revisions

Dear authors,

Many thanks for your submission. I agree with the reviewers that your MS only requires minor revision before we can accept it.

Please pay particular attention to the use of the word "basal" (see reviewer 1), the suggestions for improving the figures (see reviewer 2), and consider adding a phylogenetic analysis (see reviewer 3). I'm open to hearing your arguments for not including a phylogenetic analysis, but I do think reviewer 3 has a point in saying that your material is not an early juvenile, so a phylogenetic analysis can be of use to support your systematic assessment, even though it may not provide the definitive placement.

Note that all reviewers have annotated manuscripts available.

Looking forward to seeing your revised manuscript, and all the best,
Emanuel Tschopp

·

Basic reporting

This article is clear and unambiguous regarding grammar, although I made a few comments on typos, as noted in the pdf attached. The main problem I spotted in this article is the use of the word "basal" throughout the text. The term "basal" makes sense only within the context of a phylogenetic tree, although it is often misused as the name for any early-diverging group in the literature.

Although a solution would be to perform a phylogenetic analysis with this material, I don't think taxonomic works necessitate the extra effort of a phylogenetic analysis. My first suggestion is to choose a reference phylogeny from the literature so the reader can understand the rationale for selecting taxa involved in the comparison. This is because the taxonomic content of "basal" depends on the chosen topology. For example, my reference phylogeny of Sander et al. 2011 has some more taxa that share basal nodes with Shunosaurus and Mamenchisaurus, e.g. Patagosaurus and Barapasaurus (which are not included in the paper).

My second suggestion is to rewrite the paper so that the word "basal" is not used as a taxonomic category. It seems to me that the point of the paper is to highlight how several characters seem plesiomorphic and are shared with non-mamenchisaurid sauropods. Still, there are also characters that clearly place the specimen as a mamenchisaurid. I agree with the explanation provided in the paper: the plesiomorphic non-mamenchisaurid characters appear in juvenile mamenchisaurids before all the mamenchisaurid characters do. However, because the word basal is often interpreted as a descriptor of the nodes, the conclusion can also be read as a proposition that the juvenile Tibet sauropod shows characters that would be in the basal nodes - which is not tested. Therefore, to avoid confusion, I would suggest replacing the world basal with "early branching", "early-diverging", "non-mamenchisaurid sauropods", and so on, accordingly.

Based on the text, I think the word "basal" creates unnecessary confusion when reading the paper. The paper would benefit from comparing the Tibet sauropod against other non-mamenchisaurid sauropods between Shunosaurus and Mamenchisaurus, following a reference phylogeny and avoiding the use of basal - which should be restricted to talking about reconstruction of characters in the nodes.

Experimental design

No comment

Validity of the findings

The character regarding the ventral keel restricted to the anterior portion of the centrum in cervical vertebrae is more widespread in non-sauropod sauropodomorphs. I added further comments on this regard in the attached pdf.

·

Basic reporting

Dear authors,

It was a pleasure reviewing your paper. Kudos to you for describing fragmentary material; it is very important for the completeness of the sauropod fossil record, but often hard and arduous work.

The paper is well written and structured, and Figures are in general very good. I wonder if line drawings are an option for better viewing of the laminations on the vertebrae? I do appreciate this is time-consuming, so I would not see it as absolutely essential. Some Figures miss some markings of elements and/or laminations (particularly posterior views of cervicals) so that could maybe be added.

See the pdf for my comments, I would say that it will be easy to answer my questions and add some more info. I'd say this paper can be accepted with Minor Revisions, and I would be happy to review any future iterations of the manuscript.

Experimental design

One question I have is, can the date be more clearly defined? A lot happens with sauropods in the Early and Middle Jurassic, and so it makes a big difference whether we're in the Toarcian, the Aalenian, or the Calovian. Is it possible to get a more accurate dating, or is that work for the future?

Validity of the findings

The Discussion could merit from a look at some more Laurasian and Gondwanan Middle Jurassic sauropods, see my comments in the pdf; I hope I gave hyperlinks to most papers I refer to, so it would be easy to get the pdfs of the papers describing sauropods like Cetiosaurus, Bagualia, Patagosaurus, etc.

Reviewer 3 ·

Basic reporting

The material is well described, although the comparison was mainly made with Chinese sauropods (and using a more widespread list of taxa will improve their quality). The English can be improved but is not difficult to understand. I made some corrections and suggestions in the attached pdf

Experimental design

I think that the authors should include a phylogenetic analysis in order to better understand the position of the materials analyzed here at least its more parsimonious position, especially given that are juvenile specimens but with a high degree of morphological maturity. So, the phylogenetic results will show a position wich can be inferred as pretty similar to that of the adult specimens of the same species.

Validity of the findings

Will be improved using the results of a phylogenetic analysis, and will not take much time to do that. With these results at hand the discussion could be mucho more improved in both terms, the faunal diversity but also in the putative ontogenetic changes comparing the materials with closely related taxa.

Annotated reviews are not available for download in order to protect the identity of reviewers who chose to remain anonymous.

---

## Round 0.2 · Minor Revisions

Dear authors,

thank you for your detailed revision and rebuttal. I especially applaud the fact that you produced 3D models in the meantime, and think the addition of the phylogenetic analysis is interesting. I have very few, additional minor comments in the annotated manuscript attached below. The only major comment would be that I would like to see the majority consensus tree as a figure in the main paper instead of having them tucked away in the supplementary materials.

Many thanks in advance, and all the best for 2023!

---

## Round 0.3 · accepted · Accept

Dear authors, many thanks for addressing the last few points in detail. I have assessed your revision and think it is ready for publication.
All the best, it was a pleasure to work with you,
Emanuel Tschopp